

# Molecular characterization of two recombinant isolates of telosma mosaic virus infecting *Passiflora edulis* from Fujian Province in China

Lixue Xie[1], Fangluan Gao[2], Jianguo Shen[3], Xiaoyan Zhang[1], Shan Zheng[1], Lijie Zhang[1] and Tao Li[1]

[1] Fruit Research Institute, Fujian Academy of Agricultural Sciences, Fuzhou, China
[2] Institute of Plant Virology, Fujian Agriculture and Forestry University, Fuzhou, China
[3] Fujian Key Laboratory for Technology Research of Inspection and Quarantine, Technology Center of Fuzhou Customs District, Fuzhou, China

## ABSTRACT

Telosma mosaic virus (TeMV) is an important plant virus causing considerable economic losses to passion fruit (*Passiflora edulis*) production worldwide, including China. In this study, the complete genome sequence (excluding the poly (A) tail) of two TeMV isolates, Fuzhou and Wuyishan, were determined to be 10,050 and 10,057 nucleotides, respectively. Sequence analysis indicated that Fuzhou and Wuyishan isolates share 78–98% nucleotide and 83–99% amino acid sequence identities with two TeMV isolates of Hanoi and GX, and a proposed new potyvirus, tentatively named PasFru. Phylogenetic analysis indicated that these TeMV isolates and PasFru were clustered into a monophyletic clade with high confidences. This indicated that PasFru and the four TeMV isolates should be considered as one potyvirus species. Two recombination breakpoints were identified within the CI and NIb genes of the Fuzhou isolate, and also within the P1 gene of the Wuyishan isolate. To the best of our knowledge, this is the first report of TeMV recombinants worldwide.

## INTRODUCTION

Passion fruit (*Passiflora edulis*), originating in South America, is an important fruit crop that comprises a variety of cultivars and it is consumed globally (*Ulmer & Mac Dougal, 2004*). In China, passion fruit orchards are mainly located in the southern part of China such as Guangxi and Fujian provinces. However, the production of passion fruit is negatively affected by various plant diseases and insect pests, especially viruses (*Amata, 2011*; *Singh, 2004*). It is documented that passion fruits are susceptible to infection of more than 25 different viruses (*Baker et al., 2011*). Telosma mosaic virus (TeMV) is one of the dominant types of plant pathogens constraining sustainable development of passion fruit production.

Telosma mosaic virus is a member of the genus *Potyvirus* (*Lefkowitz et al., 2018*) and has a +ssRNA genome of about 9.7 kb (flanked by UTR at 5′ and 3′ ends) that encodes a

Corresponding authors
Fangluan Gao, Raindy@fafu.edu.cn
Tao Li, leetao06@163.com

polyprotein of 350 kDa, which is cleaved into 10 functional proteins by virus-encoded proteinases (*Lefkowitz et al., 2018*). In addition, a short ORF (PIPO) is translated by +2 nucleotide frame shifting within the P3 cistron and expressed as a P3–PIPO fusion product (*Chung et al., 2008*). TeMV was firstly reported to infect Chinese violet (*Telosma cordata*) in Vietnam (*Ha et al., 2008*), patchouli (*Pogostemon cablin*) in Indonesia (*Noveriza et al., 2012*), then passion fruit in Thailand (*Chiemsombat, Prammanee & Pipattanawong, 2014*). Recently this virus has also been reported to be present in China (*Chen et al., 2018*; *Xie et al., 2017*; *Yang et al., 2018*).

During a survey of passion orchards in 2017 in Fujian Province, China, plants exhibiting virus-like mosaic and crinkle symptoms were prevalent. The disease causes a serious reduction in production and decreases the quality of passion fruit (*Xie et al., 2017*). In 2018, a TeMV isolate named PasFru (accession number: MG944249), collected from Haikou city in Hannan Province, China, was identified and proposed to be a new member of the *Potyvirus* genus based on analyses of the complete genome sequence (*Yang et al., 2018*). In addition to PasFru, only one complete genome of TeMV isolate (named Hanoi) from Vietnam has been deposited in GenBank (accession number: NC_009742), although TeMV has been identified in many countries. One nearly complete genome of TeMV isolate from Guangxi Province in China (named GX, accession number: KJ789129) is also available in GenBank. To date, no recombinant TeMV isolate has been reported worldwide.

The objectives of this study were (i) to identify two new TeMV isolates from passion fruit in China using transmission electron microscopy, indirect ELISA and RT-PCR; (ii) to obtain their complete genome sequences and characterize their genomic structure; and (iii) to clarify the current confusion surrounding the taxonomic status of some of these TeMV isolates, particularly the proposed new potyvirus PasFru.

## MATERIALS AND METHODS

### Sample collection, electron microscopy and serological detection

Two passion fruit samples showing mosaic and crinkle symptoms on the leaves (Fig. 1A) were collected in 2017 from a commercial orchard in Fujian Province, China (*Xie et al., 2017*). After negatively staining with 2% phosphotungstic acid (pH 6.7), crude sap from the passion fruit sample was placed onto formvar-coated copper grids, and then examined using an H-7650 transmission electron microscope (Hitachi, Tokyo, Japan) operating at 80 kV. Fresh leaf samples of passion fruit were manually homogenized in pestles for homogenization with 0.05 M sodium carbonate buffer, pH 9.6. The antigen-coated indirect ELISA protocol was performed by using universal potyvirus antiserum (Agdia, Elkhart, IN, USA) according to the manufacturer's instructions. All samples were tested in duplicate wells in microtiter plates. Absorbance values at 405 nm were measured with an automatic ELISA reader (Infinite M200, Tecan, Männedorf, Switzerland). Sample with absorbance value at least twice that of healthy control was considered positive.

### RNA extraction and cDNA synthesis

Total RNA was extracted from the leaf tissue which was positive with virus infection by ELISA using an RNA extraction kit (Qiagen, Hilden, Germany). The quantity and quality

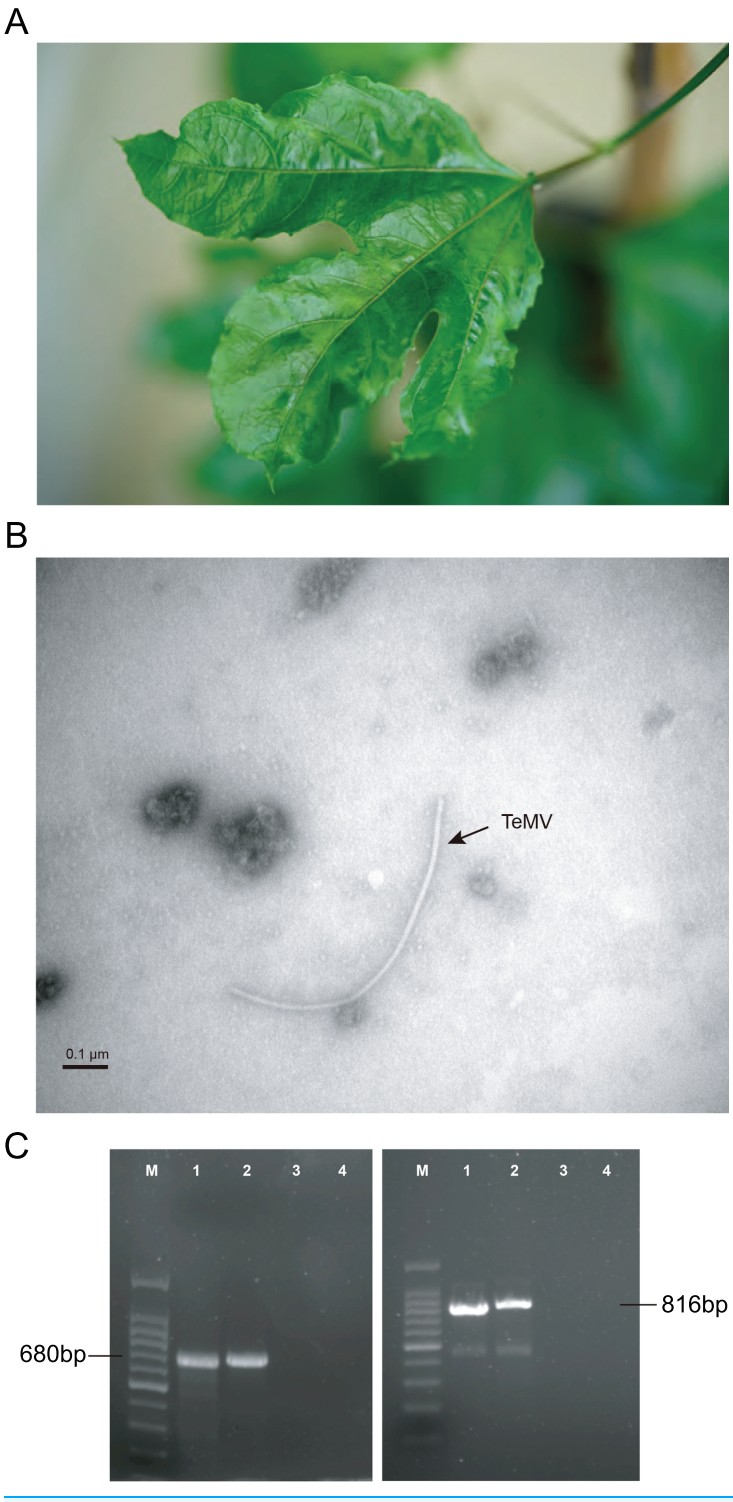

**Figure 1** **Identification of *P. edulis* leaves infected with telosma mosaic virus (TeMV).** (A) Associated disease symptoms on passion fruit infected with TeMV isolate of Fuzhou. (B) Transmission electron micrographs of virions from crude extracts of *P. edulis* infected with TeMV. (C) RT-PCR amplification of partial TeMV NIb-CP and entire CP genes, respectively. The fragments are separated in agarose gel electrophoresis. 100 bp DNA ladder (lane M). TeMV isolates of Fuzhou and Wuyishan (lanes 1–2), and negative control (lanes 3–4).               

of the extracted RNA were determined by measuring absorptions at 260–280 nm with a NanoDrop 2000c (Thermo Scientific, Waltham, MA, USA). The first-strand cDNA was synthesized using Moloney murine leukemia virus (M-MLV) reverse transcriptase (Promega, Madison, WI, USA) following the manufacturer's protocol. For the reaction, in total of 11 µl mixture containing three µl RNA (~1 µg), one µl random primer (100 µm) and seven µl DEPC-treated water were incubated at 70 °C for 10 min. Then, the mixture was transferred immediately to an ice bath for 5 min. Finally, five µl 5× buffer (Promega, Madison, WI, USA), two µl dNTP mix (Promega, Madison, WI, USA), one µl RNAsin Plus RNase inhibitor (Promega, Madison, WI, USA) and one µl M-MLV reverse transcriptase (Promega, Madison, WI, USA) were added to the mixture of primer and RNA. The RT reaction was carried out at 42 °C for 60 min followed by at 70 °C for 10 min. The cDNA was chilled on ice and stored at −70 °C.

## Molecular detection of virus using polymerase chain reaction

For the detection of the genus *Potyvirus*, a set of universal potyvirus primer LegPotyF 5′-GCWKCHATGATYGARGCHTGGG-3′ and LegPotyR 5′-AYYTGYTYMTCHCCAT CCATC-3′ (*Wylie et al., 2010*) was used to amplify a fragment of approximately 680 bp. Polymerase chain reaction (PCR) reactions were performed in a 25 µl volume containing 12.5 µl of 2× *Taq* PCR mix (Promega, Madison, WI, USA), one µl of each primer (10 µM), two µl of cDNA and 8.5 µl DEPC-treated water. PCR conditions were as follows: initial denaturation at 94 °C for 3 min, followed by 35 cycles of denaturation at 94 °C for 30 s, annealing at 45 °C for 45 s and an extension step at 72 °C for 1 min. The amplification program was followed by a final extension step at 72 °C for 10 min.

Telosma mosaic virus was detected by using specific primers TeMV-CPf 5′-TCAAGT AAGGTGGATGATGTT-3′ and TeMV-CPr 5′-CTGCACAGAGCCAACCCCAA-3′ as previously described (*Xie et al., 2017*). The primer pair TeMV-CPf/TeMV-CPr was designed to amplify the full length of the coat protein (CP) gene (~816 bp). PCR was performed in a final volume of 25 µl in a reaction mixture consisting of 2.5 µl of 10× PCR Buffer (TaKaRa, Dalian, China), 2.5 µl of $MgCl_2$ (25 mM), one µl of dNTPs (2.5 mM), one µl of each primer (10 µM), 0.2 µl of *Taq* DNA polymerase (5 U/µl), three µl of cDNA and 13.8 µl DEPC-treated water. PCR conditions were as follows: initial denaturation at 94 °C for 3 min, followed by 35 cycles of denaturation at 94 °C for 30 s, annealing at 55 °C for 1 min and an extension step at 72 °C for 1 min. A final 10 min elongation step at 72 °C was performed at the end of the 35 cycles.

## Cloning and sequencing of TeMV complete genome

Polymerase chain reaction products were analyzed by 1.5% agarose gel electrophoresis, stained with GelRed and photographed under UV-light. The target fragments of PCR products were purified by using Agarose Gel DNA Purification Kit (TaKaRa, Dalian, China). The purified products were ligated to pMD18-T vector (TaKaRa, Dalian, China) and then transformed into the *Escherichia coli* DH5α competent cell. The positive clones containing the insert fragment were identified by PCR. Three of the positive

clones were sequenced by Shanghai Sangon Biological Engineering Technology and Service Co., Ltd.

To amplify and clone the full-length genome sequences, five overlapping fragments covering the coding regions of the TeMV genome were amplified using RT-PCR with 5 pairs of specific primers designed from the highly conservative region of the TeMV genome (Table S1). RACE PCRs for the 5′ and 3′-ends of the virus genome were conducted using the SMARTer RACE 5′/3′ Kit (TaKaRa, Dalian, China) and 3′-Full RACE Core Set with PrimeScript™ RTase (TaKaRa, Dalian, China). Long PCR fragment was determined by using primer-walking method (Sangon Biotech, Shanghai, China). At least three colonies derived from each transformant were sequenced and the consensus sequences were used for genome assembling using DNAMAN version 9.0 program (Lynnon, QC, Canada). Complete genome sequences of the Fuzhou and isolate (accession number: MK340754) and Wuyishan isolate (accession number: MK340755) were deposited in GenBank.

## Phylogenetic and recombination analyses

The complete genome was assembled from overlapping RT-PCR clones after removal of the vector and primer sequence. To identify the closest relatives of the Fuzhou and Wuyishan isolates, we performed a BLASTn search against the nt/nr databases and a sequence identity matrix using BioEdit. The putative cleavage site patterns in the polyprotein were identified using online website (http://www.dpvweb.net/potycleavage/index.html).

To reveal the evolutionary relationship of TeMV, the reference sequences of other potyviruses were retrieved from the NCBI GenBank database. We aligned sequences (excluding the UTRs) by codon and removed poorly aligned regions using TranslatorX (Abascal, Zardoya & Telford, 2010). Maximum-likelihood-based phylogenetic analysis was performed using IQ-TREE 1.6.6 (Nguyen et al., 2014) under the GTR + F + R5 nucleotide substitution model, which was selected by ModelFinder (Kalyaanamoorthy et al., 2017). Topological support was estimated by 5,000 Ultrafast bootstrap replicates as well as the SH-aLRT test with 1,000 replicates (Guindon et al., 2010).

Recombination analysis were conducted using seven different methods (RDP, GENECONV, BOOTSCAN, MAXCHI, CHIMAERA, SISCAN and 3SEQ) implemented in the RDP4 package (Martin et al., 2015). For each putative recombinant breakpoint, a Bonferroni-corrected $p$-value cutoff of 0.01 was calculated. To reduce the presence of false positives, recombination events supported by at least four methods with an associated $p$-value of $< 10^{-6}$ were considered to be significant.

# RESULTS

## Detection of TeMV

The results of electron microscopy showed the presence of potyvirus-like flexuous rod particles of about 750–770 nm in length (Fig. 1B). Presence of potyvirus infection was confirmed by using indirect ELISA. Parts of the nuclear inclusion protein b (NIb) and CP

gene of TeMV (680 bp) and the entire CP gene of TeMV (816 bp) were obtained by RT-PCR (Fig. 1C).

A BLASTn search against GenBank indicated that the RT-PCR sequences obtained here share more than 98% nucleotide identities with TeMV (accession number: KJ789129). These results suggested that TeMV was present in passion fruit plants showing mosaic and crinkle leaves from Fuzhou and Wuyishan.

## Molecular genomic characterization of TeMV isolate

The complete genome sequence (excluding the 3′ poly (A) tail) of Fuzhou is 10,050 nucleotides (nts) in length and of Wuyishan is 10,057 nts in length. Their 5′-untranslated regions (5′-UTRs) are 169 and 177 nts, respectively, while the 3′-UTRs were both 251 nts (Table 1). Both contained an open reading frame of 9,630 nts, encoding a polyprotein of 3,209 amino acids. An additional short open reading frame translated by ribosomal frameshift, called PIPO, was also identified within the P3 cistron (Chung et al., 2008); this includes the highly conserved $G_2A_6$ motif at nucleotide positions 2,873–3,913 in the genome of the Fuzhou isolate, and at nucleotide positions 2,880–3,920 in the genome of the Wuyishan isolate.

The polyproteins of Fuzhou and Wuyishan were predicted to be proteolytically processed into ten mature peptides. Their cleavage sites are in consensus to those of other TeMV isolates, whose dipeptides are Y/S, G/G, Q/G, Q/S, Q/S, Q/G, E/S, Q/S and Q/S (Fig. 2). Details regarding genome organization, and protein sizes are presented in Table 1.

Pairwise comparisons showed that Fuzhou shares 78%, 98% and 96% nucleotide sequences and 84%, 99% and 98% amino acid sequence identities with Hanoi, GX and PasFru, respectively at the complete genome level (Table 1A). Wuyishan shares 78%, 99% and 88% nucleotide sequences and 83%, 99% and 92% amino acid sequence identities with Hanoi, GX and PasFru, respectively at the complete genome level (Table 1A). At the individual cistron level, Fuzhou and Wuyishan share 43–100% nucleotide sequence identity and 40–100% amino acid sequence identity with Hanoi, GX and PasFru, respectively (Table 1B). However, the CP of Fuzhou and Wuyishan both share more than 85.9% nucleotide and 88.9% amino acid sequence identities with Hanoi, GX and PasFru, respectively. According to the accepted species demarcation criterion for the genus *Potyvirus*, Fuzhou, Wuyishan, Hanoi, GX and PasFru should be considered as one species of *Potyvirus*. Interestingly, the N-terminal region of P1 sequences of Fuzhou and Wuyishan are divergent to both length and amino acid sequences, especially with Hanoi (Fig. 2B).

## Phylogenetic classification of TeMV

Our phylogenetic analysis indicated that Wuyishan and Fuzhou were clustered into a monophyletic clade with high confidence (UFBoot/BPs = 100), together with GX, Hanoi and PasFru (Fig. 3), suggesting that these isolates share a common ancestral origin. Notably, PasFru was not placed in a new taxon, although it was proposed to be a new member of the *Potyvirus* species (Yang et al., 2018). Hanoi, isolated from *T. cordata* in

**Table 1** (A) Percentage of nucleotide (below diagonal) and amino acids (above diagonal) sequence identities of the complete genome among telosma mosaic virus isolates. (B) Percentage nucleotide and amino acids (in parentheses) identities of the untranslated.

| Virus isolates | Fuzhou | Hanoi | GX | PasFru | Wuyishan |
|---|---|---|---|---|---|
| **(A)** | | | | | |
| Fuzhou | – | 84 | 99 | 98 | 93 |
| Hanoi | 78 | – | 87 | 84 | 83 |
| GX | 98 | 79 | – | 97 | 99 |
| PasFru | 96 | 78 | 92 | – | 92 |
| Wuyishan | 92 | 78 | 99 | 88 | – |

| Segment | Genome position (Fuzhou/Wuyishan) | Protein size | | Sequence identity | | |
|---|---|---|---|---|---|---|
| | | aa | kDa | GX | PasFru | Hanoi |
| **(B)** | | | | | | |
| 5′-UTR | 1–169/1–177 | – | – | – | 99/100 | 70/93 |
| P1 | 170–1,501/178–1,508 | 444 | 50 | – | 91/62.2 (90/61) | 47/43 (45/40) |
| HC-pro | 1,502–2,872/1,509–2,879 | 457 | 52 | – | 99/88 (99/94) | 74/75 (82/82) |
| P3 | 2,873–3,913/2,880–3,920 | 347 | 41 | 58/63 (59/63) | 98/90 (99/93) | 70/70 (66/66) |
| PIPO | 3,325–3,558/3,332–3,565 | 76 | 8.7 | 97/97 (95/100) | 99/99 (99/96) | 81/81 (68/71) |
| 6K1 | 3,914–4,069/4,050–4,076 | 52 | 6 | 96/100 (100/100) | 99/96 (100/100) | 79/79 (85/85) |
| CI | 4,070–5,971/4,077–5,978 | 634 | 71 | 99/100 (100/100) | 91/90 (99/99) | 80/80 (93/93) |
| 6K2 | 6,972–6,130/5,979–6,137 | 53 | 6 | 100/99 (100/100) | 84/84 (92/92) | 74/74 (85/85) |
| VPg | 6,131–6,700/6,138–6,707 | 190 | 22 | 100/100 (100/100) | 94/93 (92/92) | 75/75 (85/85) |
| NIa | 6,701–7,429/6,708–7,436 | 243 | 28 | 100/100 (100/100) | 93/93 (95/95) | 77/77 (84/84) |
| NIb | 7,430–8,980/7,437–8,987 | 517 | 60 | 98/100 (100/100) | 94/92 (96/96) | 81/81 (87/87) |
| CP | 8,981–9,796/8,988–9,803 | 272 | 31 | 96/100 (97/99) | 99/96 (100/97) | 86/86 (90/89) |
| 3′-UTR | 9,800–10,050/9,807–10,057 | – | – | 99/98 | 99/98 | 91/90 |

Vietnam, was clustered in an outer branch of the other TeMV isolates collected from *P. edulis* in China.

## Recombination signals in TeMV

Recombination analyses identified a recombination region of approximately 4,000-nucleotides in the Fuzhou genome, with two recombination breakpoints within the CI and NIb regions (Fig. 3B) and PasFru and GX isolates were identified as its parents with a high level of significance by the RDP suite (GENECONV, $p \leq 7.70\text{E} \times 10^{-134}$, Bootscan,

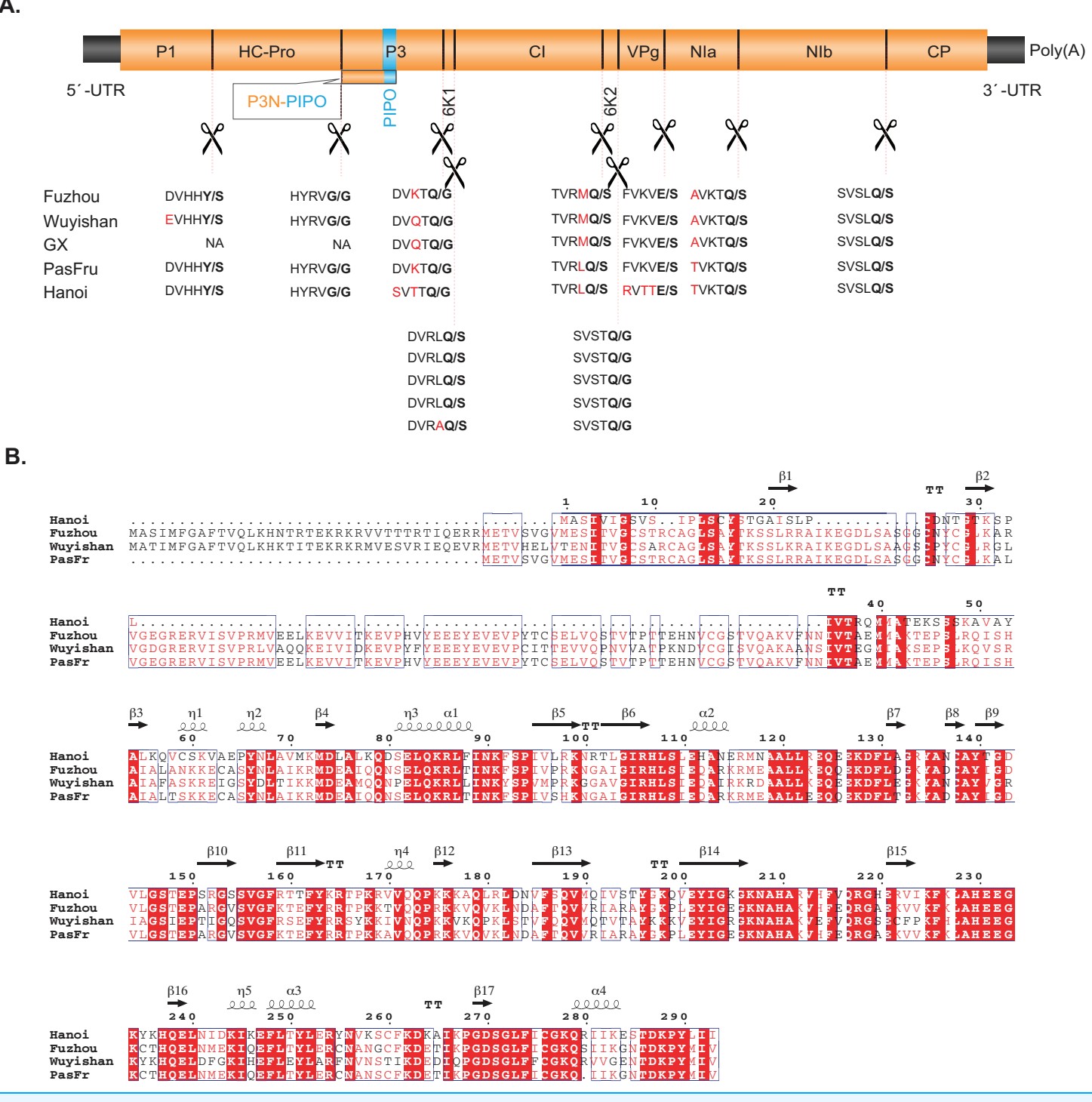

**Figure 2** **(A) Comparisons between the predicted protease cleavage sites of telosma mosaic virus (TeMV) isolates. (B) Multiple sequence alignment of the P1 protein of four TeMV isolates.** Letters in bold and in red font indicate the dipeptide cleavage sites and variable amino acid residues around the cleavage sites, respectively.

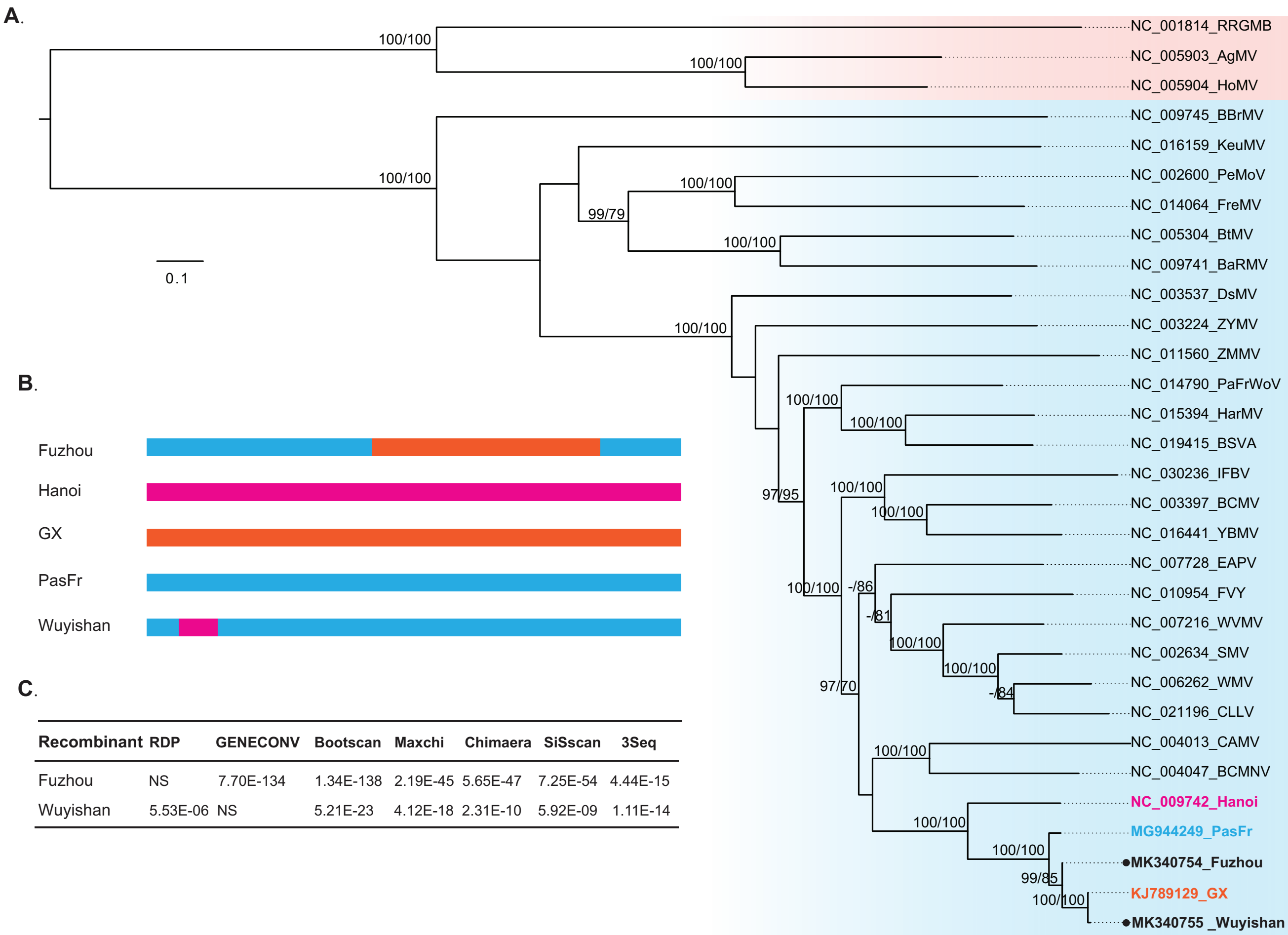

| Recombinant | RDP | GENECONV | Bootscan | Maxchi | Chimaera | SiSscan | 3Seq |
|---|---|---|---|---|---|---|---|
| Fuzhou | NS | 7.70E-134 | 1.34E-138 | 2.19E-45 | 5.65E-47 | 7.25E-54 | 4.44E-15 |
| Wuyishan | 5.53E-06 | NS | 5.21E-23 | 4.12E-18 | 2.31E-10 | 5.92E-09 | 1.11E-14 |

**Figure 3 Evolutionary relationship and recombination pattern of telosma mosaic virus (TeMV).** (A) Maximum-likelihood phylogenetic tree inferred from codon-aligned nucleotide sequences of the polyproteins of representative species of the genus *Potyvirus*, downloaded from GenBank. Three Rymoviruses (accession numbers: NC_005093, NC_005904 and NC_001814) were used as outgroups. Strong node support (UFboot support > 95% and SH-aLRT > 70%) values are shown on each node. The five TeMV isolates are indicated in bold font, and Wuyishan and Fuzhou, TeMV isolates sequenced in this study are marked by black dots. (B) Graphical representation of recombinant regions in two TeMV isolates, Fuzhou and Wuyishan. (C) Recombination detection in TeMV genomes by using seven different algorithms implemented in RDP suite. NS, not significant.

$p \leq 1.34 \times 10^{-138}$; Maxchi, $p \leq 2.19 \times 10^{-45}$; Chimaera, $p \leq 5.65 \times 10^{-47}$; Siscan, $p \leq 7.25 \times 10^{-54}$ and 3Seq, $p \leq 4.44 \times 10^{-15}$) (Fig. 3C). A short recombination region within the P1 region was identified in the Wuyishan genome (Fig. 3B) and PasFru and Hanoi isolates were recognized as its parents with high degree of support by RDP ($p \leq 5.53 \times 10^{-6}$), Bootscan ($p \leq 5.21 \times 10^{-23}$), Maxchi ($p \leq 4.12 \times 10^{-18}$), Chimaera ($p \leq 2.13 \times 10^{-10}$), Siscan ($p \leq 5.92 \times 10^{-9}$) and 3Seq ($p \leq 1.11 \times 10^{-14}$) (Fig. 3C).

## DISCUSSIONS

Recombination is an important evolutionary factor that generated genetic variation in viral populations. The phenomenon is prevalent in potyviruses, for example, in turnip mosaic virus (*Yasaka et al., 2015*), potato virus Y (*Hu et al., 2009*) and Ornithogalum mosaic virus (*Gao et al., 2018*). Among the potyviruses infecting passion fruit, TeMV has been a

prevailing virus throughout southern China in recent years (*Chen et al., 2018*; *Xie et al., 2017*; *Yang et al., 2018*). However, the recombinant TeMV isolate has not yet been investigated up to date. In the current study, the complete genomes of two isolates of TeMV were fully sequenced from passion fruit plants in Fujian, China. We provided for the first-time evidence that intra-species recombination has occurred in the genomes of Fuzhou and Wuyishan isolates. One possible explanation is that the recombinant isolates generally possess a fitness advantage over nonrecombinants (*Quenouille, Vassilakos & Moury, 2013*). Vegetatively propagated crops are particularly susceptible to viral infection (*Kraus et al., 2008*). As passion fruit cultivation relies heavily on vegetative propagation, the imported passion fruit may certainly carry a risk of introduction of passion fruit pathogens, including the new recombinant TeMV isolates, prompting us to pay more attention to their transmission in the production of passion fruit.

In the family *Potyviridae*, the species demarcation criteria for the complete ORF are less than 76% nucleotide and 82% amino acid identity. The thresholds for species demarcation are <58% nucleotide identity for the P1 coding region, and <74–78% nucleotide identity for other coding regions. However, for the CP, the optimal species demarcation criteria are <76–77% nucleotide and <80% amino acid identity, respectively. According to this criterion, Fuzhou, Wuyishan, GX, PasFru and Hanoi should be considered as isolates of TeMV. In *Potyvirus*, P1 is the most divergent protein varying in length and its amino acid sequences (*Rohožková & Navrátil, 2011*), which is considered to be the determinant of potyviruses adapting to a wide range of host species (*Valli, López-Moya & García, 2007*). *Yang et al. (2018)* found molecular differences between TeMV isolate PasFru and Hanoi, particularly in the P1 coding region. A similar observation was also made for two TeMV isolates sequenced in this study—the N-terminal region of P1 sequences of Fuzhou and Wuyishan are divergent to Hanoi both in length and amino acid sequences.

The results of this study identified PasFru as the common progenitor of Fuzhou and Wuyishan. This suggested that PasFru possibly emerged more ancient than other TeMV isolates in China. However, the temporal dynamics of TeMV could not be estimated since the TeMV isolates in our study are relatively limited, particularly the earlier isolates were not sampled. In addition, TeMV tended to cluster according to their geographical or host species origin; this could be explained, in part, as geography-specific or host-driven adaptation (*Xie et al., 2017*). However, this possibility was not tested due to the limited number of geographic region and host species available. Further studies aiming to understand the evolutionary timescale and patterns of adaptive evolution of TeMV will be interesting based on larger data sets. These will lead to a more comprehensive view of the evolutionary history of TeMV.

## CONCLUSIONS

In summary, this study represents one of serval attempts to reveal the taxonomic status of TeMV isolates. Our sequence analyses suggest that PasFru and other TeMV isolates should be considered as one potyvirus species. In addition, we found Fuzhou and

Wuyishan were TeMV recombinants. To the best of our knowledge, this is the first report of TeMV recombinants worldwide.

### Funding

This work was supported by the National Key R&D Program of China (Grant No. 2016YFF0203200), the Agricultural Science and Technology Major Project Funds of Fujian (Grant No. 2017NZ0003-1), General Project of Fujian Academy of Agricultural Sciences (Grant No. AA2018-1) and the Science and Technology Innovation Team Project of Fujian Academy of Agricultural Sciences (Grant No. STIT2017-2-4). The funders had no role in study design, data collection and analysis, decision to publish, or preparation of the manuscript.

### Grant Disclosures

The following grant information was disclosed by the authors:
National Key R&D Program of China: 2016YFF0203200.
Agricultural Science and Technology Major Project Funds of Fujian: 2017NZ0003-1.
General Project of Fujian Academy of Agricultural Sciences: AA2018-1.
The Science and Technology Innovation Team Project of Fujian Academy of Agricultural Sciences: STIT2017-2-4.

### Competing Interests

The authors declare that they have no competing interests.

### Author Contributions

- Lixue Xie conceived and designed the experiments, performed the experiments, analyzed the data, prepared figures and/or tables, and approved the final draft.
- Fangluan Gao conceived and designed the experiments, analyzed the data, prepared figures and/or tables, authored or reviewed drafts of the paper, and approved the final draft.
- Jianguo Shen conceived and designed the experiments, analyzed the data, prepared figures and/or tables, authored or reviewed drafts of the paper, and approved the final draft.
- Xiaoyan Zhang performed the experiments, prepared figures and/or tables, and approved the final draft.
- Shan Zheng performed the experiments, prepared figures and/or tables, and approved the final draft.
- Lijie Zhang performed the experiments, prepared figures and/or tables, and approved the final draft.
- Tao Li conceived and designed the experiments, analyzed the data, prepared figures and/or tables, authored or reviewed drafts of the paper, and approved the final draft.

## Ethical approval

This article does not contain any studies with human participants or animals performed by any of the authors.

## Data Availability

Sequences are available at NCBI GenBank: MK340754 and MK340755.

## Supplemental Information

Supplemental information for this article can be found online at http://dx.doi.org/10.7717/peerj.8576#supplemental-information.

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
