# Peer review of "Molecular characterization of two recombinant isolates of telosma mosaic virus infecting Passiflora edulis from Fujian Province in China"

_PeerJ, doi:10.7717/peerj.8576_

## Round 0.1 · original submission · Major Revisions

While I indicate that Major Revisions are necessary, this is mostly because of the number of suggested modifications. But overall, the reviews of the manuscript were very good, and acceptance is likely if the concerns of the reviewers are addressed. In particular, please carefully consider the critique from Reviewer #1 who provided suggestions for expanding a number of points within the manuscript including the methods used for sequencing, and especially evidence for and consequences of recombination.

·

Basic reporting

This manuscript reports on the complete genome sequences of two Telosma mosaic virus isolates isolated from Passiflora edulis plants in China, and evidence of recombination within each genome. The authors also propose that isolate PasFru, previously proposed as representing a novel species, be accepted as an isolate of Telosma mosaic virus.

Minor grammatical errors are prevalent throughout the manuscript, and I suggest the authors employ a good editing service to correct these.

Some specific points are given below:
L 39-41 It is documented that passion fruits are susceptible to infection of more than 25 different viruses (Baker et al. 2011), in which several different potyviruses have been observed worldwide.
Please delete or clarify the phrase following the comma.
L43. Provide a reference for this statement and the one in lines 54-55.
L44. Write the full name of the virus species (in italics). ‘Telosma mosaic virus is a species of the genus Potyvirus. TeMV has a positive-sense ssRNA…
L51. Which plant host was it reported from in China?
L57. Change Potyvirus species to Potyvirus genus, or a new species of potyvirus
L58-60. Please clarify this sentence. Name the Vietnamese isolate.

Experimental design

L72. Clarify what you mean by ‘randomly collected’. Were they chosen based on symptoms? Please describe the context of the infected plants: commercial orchard, home garden, feral plants etc

L71 How certain are the authors that these symptoms were induced by TeMV infection? Could other viruses be present. Were transmission experiments done to healthy passiflora or other plants? If not, please explain why not.

Line 124-125. Change 5 to five
L129. Clarify what is meant by ‘As an additional check to clarify sequence accuracy, the viral genome sequences were repeated at least three times.’

Table S1. I am uncertain how five sets of primers were sufficient to sequence the entire genome. Each amplicon would have to be >2 Kb, and to sequence each amplicon in both directions would require several other sets of primers. This section is not well -described, but it should be.

Validity of the findings

L182 remove ‘more than’ and provide numbers for each comparison.

L196. Provide the statistical evidence for recombination in this section.

L204. The Potyviridae is not the largest family of plant viruses. I would like to advise that the first small paragraph not be a repeat of the introduction, as it currently stands, but rather a one or two sentence summary of the aims and results of the study. Along the lines of ‘The complete genomes of two isolates of TeMV were fully sequenced from passion fruit plants in China. We provided for the first time evidence that intra-species recombination has occurred…” Then go on to discuss implications of your findings in terms of evolution of this species, and speculate on how this may influence pathogenicity, transmission etc, as you have done from line 212.

The paragraph from 217-228 should come after the discussion on recombination.

Table 1A. Please provide amino acid identities too for full or partial ORF too.

Fig 1A Wuyishan appears to be color-enhanced. I suggest deleting it and retaining the image Fuzhou if this represents typical symptoms. Another option is to remove these images and provide a thorough description.

L213 Provide a reference for this statement.
L216. A reference required on risks of virus accumulations in vegetatively-propagated plants.

Additional comments

The final paragraph of the Discussion is weak. Perhaps point out some of the weaknesses of the current study (only two isolates, only one host species sampled, only one location sampled, the inherent limitations of RT-PCR-based sequencing, etc) and where further studies could focus.

Your main finding is essentially two complete genomes of isolates of Telosma mosaic virus. Isolates of this species have been provided by other previously, so I suggest you place greater emphasis on the recombination finding and its implications.

Reviewer 2 ·

Basic reporting

no comment'

Experimental design

no comment'

Validity of the findings

no comment'

Additional comments

In this study, authors analyzed the complete genome sequence of two TeMV isolates, Fuzhou and Wuyishan and identified the recombination breakpoints within the CI and NIb genes of the Fuzhou isolate, also within the P1 gene of the Wuyishan isolate for the first time. There some problems should be
corrected befored pubilication.
1. The title "in china" is too large to cover you contents. because the authors only analyzed the samples collected from Fujian. so please change into Fujian.
2. in materials and methods, line 83, "RNA extraction and reverse transcriptase polymerase chain reaction (RT-PCR)", but the content afterward did not contained the RT-PCR. you should combined this part with the part of "Molecular detection of virus".
3. line 157, the results of Indirect-ELISA were not showed in the part of results. you should add the results or delete the methods in the revised manuscript.
4. TeMV-infected P. edulis showed severe symptoms, such as mosaic and distorted leaves, mosaic skin on green fruit and decreased fruit size. but the picture in figuer 1 was not typical.
5. there is a blank between numbers and unites. such as line 161. you should check it throughout the article.
6. please delete the from passion fruit in line 72.
7. please add is between it and consumed in line 36.
8 in fact, Chen et al identified Telosma mosaic virus infection in Passiflora edulis and analyzed its impact on phytochemical contents. you shuold cite the article.

Reviewer 3 ·

Basic reporting

In general the manuscript has been written with professional english. As some typos can still be found, another round of proofreading could improve the presentations. For example, in the abstract on row 27 - phylogenetic indicate (phylogenetic analysis indicates?). Also TeMV isolate PasFr is abbreviated in the abstract ParFru.

Experimental design

This reserach pertains to biological sciences. The aim of the research was to understand the nature of the viruses affecting the yield and quality of passion fruits in orchards in Fuijan province in China. The viruses within the symptomatic passion fruit plants were identified as potyviruses and by sequencing to be isolates of Telosma mosaic virus (TeMV). Therefore, the third aims was to provide more sequence information for taxonomic decisions concerning the TeMV isolates.
After initial screening of the plant samples with universal potyvirus antiserum in an indirect ELISA test, those that were regarded as positive were subjected to further analysis. The methods how the full-length sequences were obtained as well as those of phylogenetic and recombination analyses have been described in sufficient detail. To my understanding the investigations were conducted to a high technical standard. For example, seven different programs were used to determine the possible areas of recombination within Fuzhou and Wuyishan isolates and the criteria for recombination event was set as follows: event had to be supported by four of the programs with a p-value less than 10-6.

Validity of the findings

The manuscript reports sequences of two novel Telosma mosaic virus isolates. The sequences indicate that these two isolates have typical potyviral sequences. It was found that there are two regions of recombinations within these RNA sequences.

The sequences of the two novel TeMV isolates show that according to species demarcation criteria (the complete ORF are <76% nucleotide identity and <82% amino acid identity) they are TeMV isolates as is also PasFr. The thresholds for species demarcation using nucleotide identity values for the individual coding regions range from 58% for the P1 coding region to 74–78% for other regions. For the coat protein, the optimal species demarcation criterion is 76–77% nucleotide and 80% amino acid identity (Adams et al., 2005b). The authors suggest that all sequenced TeMV isolated should be considered as one potyvirus species. Except the high variation in P1 gene of Hanoi isolate this statement is well supported by the sequences.

---

## Round 0.2 · Minor Revisions

Thank-you for submitting the revised version of your manuscript. As you can see from the reviews below, only a few minor corrections are needed before your manuscript can be accepted for publication. As soon as you submit a revised copy, I will be happy to officially accept your manuscript.

·

Basic reporting

Acceptable

Experimental design

Acceptable

Validity of the findings

Acceptable

Additional comments

I have read the rebuttal and the manuscript and I am satisfied that the authors have made a thorough job of amending the manuscript. I feel that it can be accepted for publication, with only minor editorial amendments required

Reviewer 2 ·

Basic reporting

no comment

Experimental design

no comment

Validity of the findings

no comment

Additional comments

In the revised manuscript, authors have revised the article as suggested by the reviewer. We agree with the publication of the article.

Reviewer 3 ·

Basic reporting

-

Experimental design

-

Validity of the findings

-

Additional comments

The revised version of the manuscript ”Molecular characterization of two recombinant isolates of telosma mosaic virus infecting Passiflora edulis from Fujian Province in China (#41501) has answered most of the reviewer’s comments. I still spotted few corrections, which could be considered:
- Introduction , row 56: only one complete TeMV isolate
- Materials and methods of cDNA synthesis and PCR: giving the volumes how much RNA/DNA and primers were pipetted into the reactions does not give an idea of the real amounts of these. Please, provide mass or volume + concentration of them.
- Although the sentence on rows 129-130 has been reformulated, it still could be made more clear: was the whole sequencing process repeated three times? Did you start from new RNA isolation, new cDNA synthesis or from the PCRs – how independent were the sequenced samples from each other?
- Row 164: Please, give the Genebank number for the TeMV genome sequence that was used here for comparison.
- Row 165-166: I think the results show that the symptomatic trees were infected by TeMV, but it is not clear whether the symptoms were caused by this virus.
- Row 174: Is the ribosomal frameshift signal G2A6 required for frameshift if its origin is transcriptional slippage?
- Figure 1 legend: The legend still tells that it shows symptoms in P. edulis infected with Fuzhou and Wuyishan isolates. Only one is left in revised Fig. 1 , which one?

---

## Round 0.3 · accepted · Accept

Thank-you for your resubmission of a revised manuscript based on the reviewer's comments. I am now happy to accept your manuscript for publication.